# REACTIONLLM: DATA-CENTRIC LEARNING FOR A COMPACT SINGLE-STEP CHEMICAL REACTION PREDICTION MODEL

## ABSTRACT

Chemical reaction prediction faces three fundamental challenges that limit practical deployment: (1) ineffective molecular representations that fail to capture essential chemical context, (2) unfair comparison with test-time augmentation and without mention the usage of AAM(atom-atom mapping, and (3) unsatisfactory performance of large-scale pretrained models. To address these limitations, we present a unified framework that enables a compact 0.5B parameter model to outperform significantly larger counterparts (7B/13B parameters) through three strategic innovations: the AAM-0 molecular representation that bridges mapped and unmapped data via implicit contrastive learning; bidirectional multi-task learning that creates a unified chemical representation space across retrosynthesis and forward prediction tasks; and structured plan-based reasoning that ensures chemically plausible step-by-step rationalizations. Extensive evaluation with rigorous separate assessment of mapped and unmapped performance demonstrates +14% accuracy improvement over strong baselines, establishing that carefully designed compact models with built-in chemical intelligence can surpass larger, less specialized alternatives while maintaining computational efficiency.The implementation is available at: `https://anonymous.4open.science/r/ReactionLLM-DF4C`.

## 1 INTRODUCTION

Chemical reaction prediction represents a fundamental challenge in computational chemistry and drug discovery, with significant implications for accelerating molecular design and synthetic planning (Long et al., 2025). The core objective encompasses two complementary tasks: *forward reaction prediction* (Schwaller et al., 2019) (determining products from given reactants) and *retrosynthetic analysis* (Corey & Wipke, 1969) (identifying plausible reactant pathways from target products). Traditional approaches (Corey & Wipke, 1969; Szymkuć et al., 2016; Bøgevig et al., 2015) rely heavily on expert-curated reaction rules and experimental trial-and-error, which are inherently limited in scalability, generalizability, and efficiency (Szymkuć et al., 2016).

The advent of artificial intelligence (AI) and deep learning has introduced data-driven paradigms (Wang et al., 2021; Coley et al., 2017a; Segler & Waller, 2017) that learn reaction patterns directly from large-scale reaction databases (Jacob & Lapkin, 2018; Kim et al., 2025). Current methodologies can be categorized into three principal paradigms: *Template-based methods* (Dai et al., 2019; Chen & Jung, 2021; Xie et al., 2023) utilize predefined reaction templates—explicit rules encoding atomic and bonding changes at reaction centers. While offering high accuracy for known reaction types and inherent interpretability (Wu et al., 2018), these approaches suffer from limited coverage of chemical space, inability to handle novel reactions beyond their template library (Coley et al., 2017b), and combinatorial explosion of required templates (Shee et al., 2024). *Semi-template methods* (Somnath et al., 2021; Zhong et al., 2023) strike a balance between template specificity and flexibility by employing generalized patterns that capture essential reaction features without exact atom-level specification. These methods mitigate the template explosion problem while retaining some interpretability, but still require significant manual curation and struggle with complex multi-step transformations (Zhong et al., 2023). *Template-free methods* (Sacha et al., 2021; Wan et al., 2022; Seo et al., 2021; Han et al., 2024) represent the most flexible approach, treating

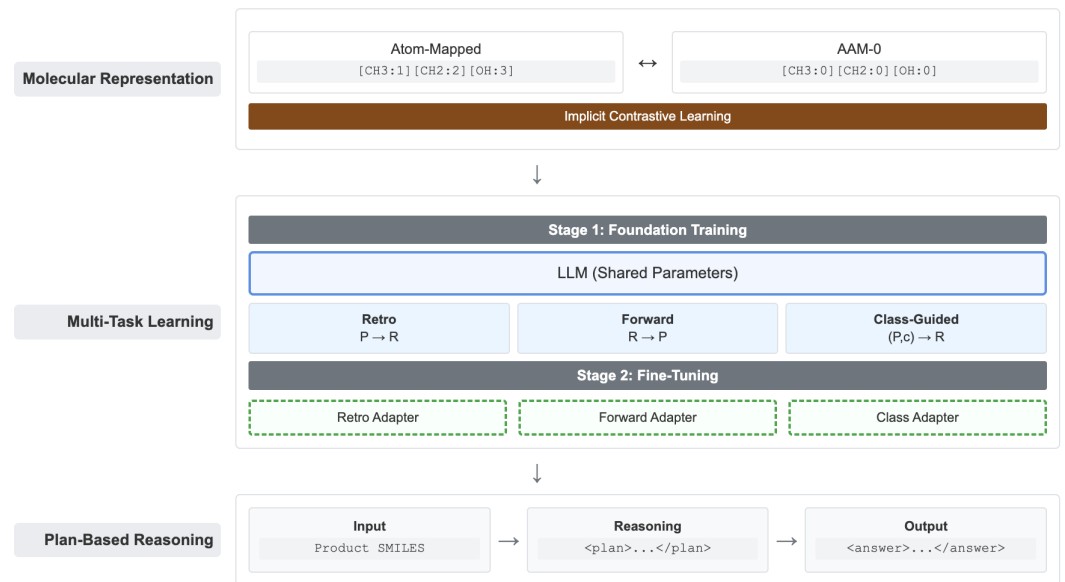

Figure 1: Overview of our unified framework for chemical reaction prediction, featuring three core components: AAM-0 molecular representation with implicit contrastive learning, bidirectional multi-task learning with two-stage training, and plan-based reasoning.

reaction prediction as a sequence-to-sequence or graph-to-graph translation problem without explicit reaction rules. While offering greater generality and can potentially discover novel reactions, they face significant challenges including: (1) tendency to generate chemically implausible ("hallucinated") products that violate fundamental chemical constraints (Schwaller et al., 2018), (2) high data requirements due to lack of inductive biases (Han et al., 2024), and (3) limited interpretability of model decisions (Zeng et al., 2024). Based on model architecture, methods can be categorized as: *Graph-based models* (Chen et al., 2019; Dai et al., 2019; Chen et al., 2023) using graph neural networks (GNNs) that operate directly on molecular graphs *Sequence-based models* (Liu et al., 2017; Karpov et al., 2019; Zheng et al., 2019) using Transformer architectures (e.g., Molecular Transformer) that treat SMILES strings as chemical "language". *Large language model (Zhao et al., 2025; 2024; Yu et al., 2024) (LLM) approaches* that leverage pretrained language models fine-tuned on chemical data, treating reactions as text generation problems.

Despite considerable advances, current approaches face three limitations:

First, ineffective molecular representations undermine model performance across all methodological paradigms. Template-based methods (Chen & Jung, 2021; Xie et al., 2023) suffer from limited coverage and inability to handle novel transformations. Semi-template approaches (Shi et al., 2020; Zhong et al., 2023) struggle with error propagation in reaction center identification. Template-free methods (Seo et al., 2021; Han et al., 2024), including LLM-based approaches (Zhao et al., 2025), often generate chemically invalid molecules due to poor representation of reaction context. The fundamental issue spans all architectures: graph-based models require complex processing, sequence-based models suffer from SMILES non-uniqueness and fragility, and molecular fingerprints lose critical structural information.

Second, a critical evaluation discrepancy exits in current literature. Models do not mention explicitly the usage of atom-atom mapping(AAM). This creates an unfair comparison paradigm, as AAM provides unique reaction signals. Some models (Han et al., 2024; Zhang et al., 2025b) use (20 times) test-time augmentation on both train and test set to enhance the evaluation performance but this gain is orthogonal to the chemical understanding. Since this means models are trained and evaluated on synthetic dataset.

Third, performance limitations of current approaches reveal fundamental gaps in chemical understanding. Large-scale models (Zhao et al., 2025; Taylor et al., 2022) achieve poor accuracy (e.g., 17.9% top-1 on USPTO-50k) despite massive parameter counts and extensive pretraining. Even spe-

cialized smaller models (Han et al., 2024; Schwaller et al., 2019) report unsatisfactory performance, suggesting inherent limitations in current architectural paradigms. The performance gap highlights that scale alone cannot compensate for lack of domain-specific chemical intelligence.

To address these challenges, we present a unified framework that integrates three innovations: **AAM-0 representation** solves the representation gap by creating a canonical format that preserves reaction center topology while enabling implicit contrastive learning between mapped and unmapped data. This approach bridges the evaluation discrepancy by ensuring fair comparison across data conditions. **Bidirectional multi-task learning** addresses performance limitations by simultaneously training on complementary tasks (retrosynthesis, class-conditioned retrosynthesis, and forward prediction), creating a unified chemical representation space that captures fundamental reactivity principles beyond pattern memorization. **Structured plan-based reasoning** uses explicit step-by-step rationalizations, enhance the two stage training without requiring annotated chain-of-thought data.

Our approach demonstrates that a compact 0.5B parameter model, when properly designed with chemical intelligence built into its architecture, can outperform significantly larger counterparts (7B/13B parameters) while maintaining computational efficiency. Extensive evaluation with rigorous separate assessment of mapped and unmapped performance shows +14% accuracy improvement over strong baselines, establishing a new paradigm for efficient and effective chemical AI.

Our work makes the following contributions:

- A novel AAM-0 representation that resolves the evaluation discrepancy between mapped and unmapped molecular data

- A bidirectional multi-task learning framework that creates a unified chemical representation space through synergistic task augmentation

- A plan-based reasoning mechanism that enhances both accuracy and interpretability while ensuring chemical plausibility

- Rigorous evaluation demonstrating that strategic architectural design can surpass scale-based approaches

## 2 RELATED WORK

**Large Language Models in Chemistry**   Large Language Models (LLMs) have been applied to chemistry with varying success (Zhang et al., 2024). While large-scale foundation models like Galactica (120B) (Taylor et al., 2022) and ChemDFM (13B) (Zhao et al., 2025) excel at general chemical question-answering tasks. These models leverage extensive pretraining on scientific corpora (Zhang et al., 2024) and advanced techniques like knowledge distillation (Zhang et al., 2025b), but their computational demands limit accessibility (Zhang et al., 2025a) and they demonstrate poor performance on specialized tasks like retrosynthesis, achieving only 17.9% top-1 accuracy on USPTO-50k (Zhao et al., 2025).

**Reaction Prediction Paradigms**   Reaction prediction includes both forward and retrosynthesis, though most research focuses on the latter (Long et al., 2025). Specialized smaller models (Schwaller et al., 2019; Irwin et al., 2022; Han et al., 2024) achieve better retrosynthesis performance. Current methodologies fall into three categories. Template-based methods (Chen & Jung, 2021; Xie et al., 2023) match reactions against predefined libraries but cannot generalize to novel transformations. Semi-template approaches (Shi et al., 2020; Zhong et al., 2023) identify reaction centers then complete synthons, but suffer from error propagation. Template-free methods (Seo et al., 2021; Han et al., 2024) treat prediction as sequence translation but need deeper understanding to chemical data. Some approaches (Han et al., 2024; Zhang et al., 2025b) use test-time augmentation to boost performance (Ramos et al., 2025), which does not improving fundamental understanding. Critically, many methods use atom-mapped (AAM) data during training but are evaluated without explicit comparison to unmapped baselines, creating an unfair performance advantage.

## 3 METHODOLOGY

### 3.1 OVERVIEW

Our methodological framework is designed to overcome the fundamental limitations of existing approaches to chemical reaction prediction. We posit that superior performance is achieved not through scale but through strategic architectural innovations that enhance data efficiency, representation learning, and reasoning fidelity. Our integrated approach, depicted in Figure 1, combines three core components: (1) a novel molecular representation (AAM-0) that bridges the gap between mapped and unmapped data formats, (2) a bidirectional multi-task learning strategy that creates a unified chemical representation space, and (3) a plan-based reasoning mechanism that provides explicit, step-by-step rationalizations for model predictions.

### 3.2 CORE PROBLEM FORMULATION

We frame chemical reaction prediction as a sequence-to-sequence transduction problem. The standard Simplified Molecular Input Line Entry System (SMILES) representation, while compact, presents fundamental limitations for this task: Non-Uniqueness: A single molecule can have multiple valid SMILES strings, introducing unnecessary variability. Lack of Reaction Context: Standard SMILES lacks explicit annotation of reaction centers, forcing the model to infer atomic correspondence. Sensitivity to Small Changes: Minor syntactic variations can alter molecular meaning, making the learning problem brittle.

Atom-mapped SMILES (AAM) addresses the second point by providing explicit atomic correspondence between reactants and products, offering a "reaction grammar." However, this detailed mapping is often unavailable during inference in real-world scenarios, creating a train-test distribution mismatch.

Let $\mathcal{X}$ and $\mathcal{Y}$ denote the input and output spaces. The objective is to learn a hypothesis $h : \mathcal{X} \to \mathcal{Y}$ that approximates $P(y|x)$ for $(x, y) \sim \mathcal{D}$.

### 3.3 AAM-0: AN EQUIVARIANT REPRESENTATION AND IMPLICIT CONTRASTIVE LEARNING

To bridge the gap between the information-rich AAM and the more common unmapped SMILES, we define a canonical transformation.

**Definition 3.1** (AAM-0 Transformation). Let $S$ be a mapped SMILES string. The function $f : \Sigma^* \to \Sigma^*$ is defined as:

$$S_{\text{AAM-0}} = f(S) = (g(t_1), g(t_2), \dots, g(t_n)) \quad \text{where} \quad g(t_i) = \begin{cases} 0 & \text{if } t_i \text{ is an AAM number} \\ t_i & \text{otherwise.} \end{cases}$$

This filters semantically null information (specific integer values) while preserving the vital structural cue of the existence of a mapping.

**Proposition 3.1** (Implicit In-Context Contrastive Learning). The pair $(S, f(S))$ for any atom-mapped SMILES string $S$ forms a *positive pair* for a contrastive objective. The model learns a latent representation $\phi(\cdot)$ such that:

$$d(\phi(S), \phi(f(S))) \ll d(\phi(S), \phi(S'))$$

for a negative example $S'$ (a different molecule), where $d$ is a distance metric. This forces $\phi$ to become invariant to the presence or absence of specific mapping numbers, learning the underlying reaction topology. This implicit objective enhances the model's robustness and generalization to unmapped inputs during testing.

**Hypothesis 3.1** (H1: Unified Representation Learning). The AAM-0 transformation $f$ creates a unified representation space. Training on $S, f(S)$ pairs allows the model to perform accurately on both mapped ($S$) and unmapped ($S_{\text{unmapped}} \approx f(S)$) inputs, effectively solving the train-test distribution gap.

### 3.4 BIDIRECTIONAL MULTI-TASK LEARNING AS LATENT SPACE REGULARIZATION

We define our training paradigm around three core tasks, explicitly categorized by their input structure:

**Reaction type unknown:** $\mathcal{T}_1$: Retrosynthesis ($P \rightarrow R$), $\mathcal{T}_2$: Forward Prediction ($R \rightarrow P$)

**Reaction type known:** $\mathcal{T}_3$: Class-guided Retrosynthesis ($P, c \rightarrow R$)

This structured approach constitutes a powerful form of *task augmentation*. The model parameters $\theta$ are decomposed into shared parameters $\theta_s$ (which compute a latent representation $\phi(x)$) and task-specific parameters $\{\theta_1, \theta_2, \theta_3\}$.

Our training regimen follows a two-stage process:

**Stage 1: Multi-Task Foundation Learning**
We first learn shared parameters $\theta_s$ that compute a unified latent representation $\phi(x)$ across all tasks:

$$\min_{\theta_s} \sum_{k=1}^{3} \mathbb{E}_{(x,y) \sim \mathcal{D}_k} \left[ \mathcal{L}_k(h(x; \theta_s, \theta_k), y) \right] \tag{1}$$

where $\mathcal{L}_k$ is the task-specific loss function. This multi-task objective encourages $\phi(x)$ to form a structured representation that captures fundamental chemical reactivity principles, with the bidirectional relationship between $\mathcal{T}_1$ and $\mathcal{T}_3$ imposing a cycle-consistency constraint that serves as a powerful regularizer.

This framework forces $\phi(x)$ to form a unified representation of chemical reactivity. The class-unconditioned tasks ($\mathcal{T}_1, \mathcal{T}_2$) teach the model fundamental reaction rules, while the class-conditioned task ($\mathcal{T}_3$) shows how to apply this knowledge under specific constraints. The bidirectional nature (forward and retro) imposes an implicit cycle-consistency constraint, acting as a powerful regularizer.

**Hypothesis 3.2** (H2: Structured Latent Manifold). The multi-task objective learns a latent representation $\phi(x)$ whose topology reflects the true reaction manifold. The subsequent task-specific fine-tuning learns lightweight functions $g_k(\phi(x); \theta_k)$ on this well-structured space, leading to superior generalization.

**Stage 2: Parameter-Efficient Fine-Tuning**
We employ Low-Rank Adaptation (LoRA) for task-specific specialization while keeping the foundational representation $\phi(x)$ frozen:

$$\Delta W = BA^T \quad \text{where} \quad B \in \mathbb{R}^{d \times r}, A \in \mathbb{R}^{r \times k}, r \ll \min(d, k) \tag{2}$$

This approach ensures that the knowledge acquired during multi-task learning is preserved while allowing for efficient adaptation to each specific task.

## 3.5 Plan-Based Reasoning as Variational Inference

We introduce a latent variable $z$ (the plan) to formalize step-by-step reasoning. The true marginal likelihood $P(y|x)$ is intractable; thus, we maximize the evidence lower bound (ELBO):

$$\log P(y|x) \geq \mathbb{E}_{z \sim Q(z|x,y)} \left[ \log P(y|x,z) \right] - D_{\text{KL}} \left[ Q(z|x,y) \, \| \, P(z|x) \right], \tag{3}$$

where $P(z|x)$ is the prior, $P(y|x,z)$ is the likelihood, and $Q(z|x,y)$ is the variational posterior.

**Proposition 3.2** (Deterministic Posterior and Implicit Augmentation). Using a deterministic posterior $Q(z|x,y) = \delta(z - z^*)$ simplifies the objective to:

$$\log P(y|x, z^*) - D_{\text{KL}} \left[ \delta(z - z^*) \, \| \, P(z|x) \right].$$

The KL term forces the prior $P(z|x)$ to align with correct chemical reasoning. During training, the model learns to generate structured plans enclosed within $\langle \text{plan} \rangle$ and $\langle /\text{plan} \rangle$ tokens, followed by the final prediction within $\langle \text{answer} \rangle$ tokens. Furthermore, each $(x, y)$ pair is effectively augmented into $(x, z^*, y)$, providing a richer learning signal from a single data point and guiding the model toward chemically plausible pathways.

## 3.6 Integrated Framework and Generalization

The synergy of all components yields the full objective:

$$\min_{\theta} \sum_{k=1}^{3} \mathbb{E}_{(x,y,z^*) \sim \mathcal{D}_k} \left[ -\lambda \log P(z^*|f(x); \theta) - \log P(y|f(x), z^*; \theta) \right], \tag{4}$$

where $\lambda$ balances the plan prediction and final answer losses. The evaluation protocol must rigorously separate performance on **mapped (AAM)** and **unmapped** test sets to accurately measure the model's ability to generalize to real-world scenarios where AAM is unavailable.

Under the assumptions that (1) $f$ is information-preserving, (2) the tasks $\{\mathcal{T}_1, \mathcal{T}_2, \mathcal{T}_3\}$ are related through a shared latent structure, and (3) the plan space $z$ captures necessary reasoning steps, the proposed framework achieves a lower expected generalization error on both mapped and unmapped inputs than models trained with standard objectives. The benefit arises from: (i) reduced effective input complexity via the AAM-0 transformation and implicit contrastive learning, (ii) representational regularization via multi-task learning on a bidirectional task graph, and (iii) decomposed complexity via plan-based reasoning, which collectively lower the complexity of the hypothesis space.

### 3.7 PROMPT FORMULATION

Here is the simplified prompt template for the tasks. The full version is in A.3.

---

**System Prompt**

Respond in the following format:
```
<plan>
...
</plan>
<answer>
```
Provide the final answer in JSON format as specified in the instruction.
```
</answer>
```

---

**`<plan>` for retrosynthesis**

```
</plan>
```
To predict the reactants for the product SMILES:
1. Identify key functional groups and structural features in the product.
2. Propose retrosynthetic disconnections based on common reaction types (e.g., esterification, amide formation, sulfonamide formation, heterocycle synthesis).
3. Validate that the proposed reactants are chemically feasible and can form the product under standard conditions.
```
</plan>
```

---

**Retrosynthesis Prompt template**

Task: Retrosynthesis
Given the product SMILES: "product"
Predict the reactants required to synthesize this product.
### Instruction:
...
- Return the predicted reactants in SMILES format as a JSON object:
"reactants": "SMILES_string".

---

## 4 EXPERIMENT

### 4.1 EXPERIMENT SETTINGS

**Datasets and baselines** For the template-based methods: RetroSim (Coley et al., 2017b), NeuralSym (Segler & Waller, 2017), GLN (Dai et al., 2019), LocalRetro (Chen & Jung, 2021), and RetroKNN (Xie et al., 2023). On the semitemplate-based methods: G2G (Shi et al., 2020), RetroXpert (Yan et al., 2020), RetroPrime (Wang et al., 2021), GraphRetro (Somnath et al., 2020), and Graph2Edits (Zhong et al., 2023). For the template-free methods: Seq2Seq (Schwaller et al., 2018), SCROP (Zheng et al., 2019), AutoSynRoute (Lin et al., 2020), GET (Mao et al., 2021), DMP fu-

sion (Zhu et al., 2023), Tied Transformer (Kim et al., 2021), MEGAN (Sacha et al., 2021), Augmented Transformer (Tetko et al., 2020), GTA (Seo et al., 2021), Graph2SMILES (D-GCN) (Tu & Coley, 2022), Retroformer (Wan et al., 2022), Chemformer (Irwin et al., 2022), Ualign (Zeng et al., 2024), R-SMILES (Zhong et al., 2022), RetroExplainer (Wang et al., 2023), EditRetro (Han et al., 2024) . The detail dataset statistics is shown in table 4, we evaluate on USPTO variants with different size the feature including USPTO-50K, 480K,FULL (Dai et al., 2019).

Table 1: Retrosynthesis Results on USPTO-50k with and without Reaction Type

| Methods | Without Reaction Type | | | | With Reaction Type | | | | Feat./Tech. |
|---|---|---|---|---|---|---|---|---|---|
| | 1 | 3 | 5 | 10 | 1 | 3 | 5 | 10 | templ/map/test aug |
| **Template-based** | | | | | | | | | |
| RetroSim | 37.3 | 54.7 | 63.3 | 74.1 | 52.9 | 73.8 | 81.2 | 88.1 | ✓/-/- |
| NeuralSym | 44.4 | 65.3 | 72.4 | 78.9 | 55.3 | 76.0 | 81.4 | 85.1 | ✓/-/- |
| GLN | 52.5 | 69.0 | 75.6 | 83.7 | 64.2 | 79.1 | 85.2 | 90.0 | ✓/✓/× |
| LocalRetro | 53.4 | 77.5 | 85.9 | 92.4 | 63.9 | 86.8 | 92.4 | 96.3 | ✓/✓/× |
| RetroKNN | 57.2 | 78.9 | 86.4 | 92.7 | 66.7 | 88.2 | 93.6 | 96.6 | ✓/-/- |
| **Semitemplate-based** | | | | | | | | | |
| G2G | 48.9 | 67.6 | 72.5 | 75.5 | 61.0 | 81.3 | 86.0 | 88.7 | ×/✓/× |
| RetroXpert | 50.4 | 61.1 | 62.3 | 63.4 | 62.1 | 75.8 | 78.5 | 80.9 | ×/✓/✓ |
| RetroPrime | 51.4 | 70.8 | 74.0 | 76.1 | 64.8 | 81.6 | 85.0 | 86.9 | ×/✓/✓ |
| GraphRetro | 53.7 | 68.3 | 72.2 | 75.5 | 63.9 | 81.5 | 85.2 | 88.1 | ×/✓/× |
| Graph2Edits | 55.1 | 77.3 | 83.4 | 89.4 | 67.1 | 87.5 | 91.5 | 93.8 | ✓/-/- |
| **Template-free** | | | | | | | | | |
| Seq2Seq | 37.4 | 52.4 | 57.0 | 61.7 | 37.4 | 52.4 | 57.0 | 61.7 | ×/-/- |
| SCROP | 43.7 | 60.0 | 65.2 | 68.7 | 59.0 | 74.8 | 78.1 | 81.1 | ×/×/× |
| AutoSynRoute | 43.1 | 64.6 | 71.8 | 78.7 | - | - | - | - | ×/×/× |
| GET | 44.9 | 58.8 | 62.4 | 65.9 | - | - | - | - | ×/×/× |
| DMP fusion | 46.1 | 65.2 | 70.4 | 74.3 | - | - | - | - | ×/×/× |
| Tied Transformer | 47.1 | 67.2 | 73.5 | 78.5 | - | - | - | - | ×/×/× |
| MEGAN | 48.1 | 70.7 | 78.4 | 86.1 | 60.7 | 82.0 | 87.5 | 91.6 | ×/✓/× |
| Augmented Transformer | 48.3 | - | 73.4 | 77.4 | - | - | - | - | ×/×/✓ |
| GTA | 51.1 | 67.6 | 74.8 | 81.6 | - | - | - | - | ×/✓/✓ |
| Graph2SMILES (D-GCN) | 52.9 | 66.5 | 70.0 | 72.9 | - | - | - | - | ×/×/× |
| Retroformer | 53.2 | 71.1 | 76.6 | 82.1 | 64.0 | 82.5 | 86.7 | 90.2 | ×/-/- |
| Chemformer | 54.3 | - | 62.3 | 63.0 | - | - | - | - | ×/×/✓ |
| Ualign | 53.5 | 77.3 | 84.6 | 90.5 | 66.4 | 86.7 | 91.5 | 95.0 | ×/×/✓ |
| R-SMILES | 56.3 | 79.2 | 86.2 | 91.0 | - | - | - | - | ×/×/✓ |
| RetroExplainer | 57.7 | 79.2 | 84.8 | 91.4 | 66.8 | 88.0 | 92.5 | 95.8 | ×/-/✓ |
| RetroDFM-R-7B | 59.0 | - | - | - | - | - | - | - | ×/✓/× |
| EditRetro | 60.8 | 80.6 | 86.0 | 90.3 | - | - | - | - | ×/✓/✓ |
| ReactionLLM-0.5B(Ours) | 69.5 | 88.4 | 92.5 | 95.3 | 71.8 | 89.0 | 93.2 | 95.9 | ×/×/× |
| **ReactionLLM-0.5B(Ours)** | **74.8** | **90.1** | **93.2** | **96.3** | **75.4** | **90.8** | **94.1** | **96.8** | ×/✓/× |

**Implementation Details** We implement our framework using the Qwen2.5-0.5B architecture as our base model. We use the AdamW optimizer with a learning rate of $1 \times 10^{-4}$, $\beta_1 = 0.9$, $\beta_2 = 0.999$, and weight decay of 0.01. Training employs a linear learning rate scheduler with warmup over 10% of the total training steps. For the LoRA components, we use a rank $r = 64$ and $\alpha = 128$, applying adapters to the query, key, value, and output projections in attention layers, as well as the gate and up/down projections in feed-forward layers. All training can be down within 24GB memory GPU.

**Evaluation** To ensure fair and comprehensive evaluation, we rigorously separate performance on atom-mapped and unmapped test sets. For each task $\mathcal{T}_k$, we evaluate: Acc@$k = \frac{1}{|\mathcal{D}_{\text{test}}|} \sum_{(x,y) \in \mathcal{D}_{\text{test}}} \mathbb{I}[\hat{y} \in \text{top-}k(y)]$ where $\hat{y}$ is the model's prediction and $\mathbb{I}$ is the indicator function. We report separate results for mapped ($\mathcal{D}_{\text{test}}^{\text{mapped}}$) and unmapped ($\mathcal{D}_{\text{test}}^{\text{unmapped}}$) evaluation sets to thoroughly assess generalization capability across different data formats.

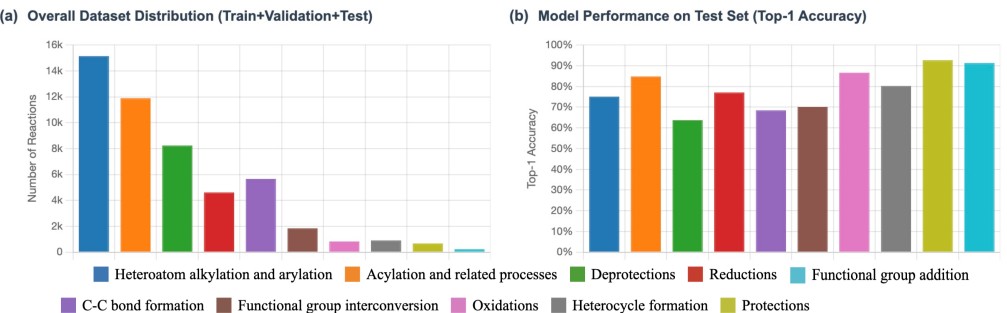

Figure 2: Statistics for USPTO-50K with different reaction types and the performances.

## 4.2 MAIN RESULTS

Table 2: Retrosynthesis and Forward Prediction Results on USPTO-full and USPTO-480k Datasets

| Methods | USPTO-full (retrosynthesis) | | | | USPTO-480k (forward prediction) | | | |
|---|---|---|---|---|---|---|---|---|
| | Top-1 | Top-3 | Top-5 | Top-10 | Top-1 | Top-3 | Top-5 | Top-10 |
| Transformer baseline | 42.9 | – | – | 66.8 | – | – | – | – |
| Molecular Transformer | – | – | – | – | 88.6 | 93.5 | 94.2 | 94.9 |
| MEGAN | – | – | – | – | 86.3 | 92.4 | 94.0 | 95.4 |
| Chemformer | – | – | – | – | 91.3 | – | 93.7 | 94.0 |
| Graph2SMILES | 45.7 | – | – | 62.9 | 90.3 | 94.0 | 94.6 | 95.3 |
| RXNGraphformer | 47.4 | 63.0 | 67.4 | 71.6 | 90.6 | 94.3 | 94.9 | 95.5 |
| RetroDFM-R-7B | 50.5 | 67.6 | 72.7 | 77.5 | - | - | - | - |
| **ReactionLLM-0.5B(Ours)** | **61.6** | **65.6** | **71.3** | **76.1** | **94.2** | **95.5** | **96.7** | **97.7** |

**Performance on USPTO-50k**   The results in Table 1 demonstrate that our method achieves state-of-the-art performance on the USPTO-50k benchmark. Our model obtains **74.8%** top-1 accuracy in type-unknown and **75.4%** in type-known scenarios, representing improvements of +14.0% and +8.7% over the best existing methods, respectively. The minimal performance difference between type-unknown and type-known conditions (0.6 percentage points) indicates effective internalization of reaction type information during training. This contrasts with traditional methods that show significant degradation without type guidance.

**Techniques used**   Moreover, methods with **test time augmentation** often conduct 20 times augmentation in both training and testing phase to improve the performance. However, the model is evaluated on more synthetic dataset in this case and lead to unfair comparison. The introducing of AAM contain more reaction information but models with AAM does not consistently outperform others. This suggest training strategy is important even with high quality data. Ours without AAM and test time augmentation still achieves sota performance compared with all the baselines.

**Scalability to Large-Scale Datasets and forward prediction**   Table 2 demonstrates our method's scalability. On USPTO-full (810k examples), we achieve **61.6%** top-1 accuracy, outperforming RXNGraphformer by +14.2%. On USPTO-480k forward prediction, we obtain **94.2%** top-1 accuracy, surpassing Molecular Transformer by +5.6%. The consistent performance gains across dataset sizes indicate that our advantages scale with data volume. The bidirectional multi-task learning approach enables effective knowledge transfer between retrosynthesis and forward prediction tasks, with each task informing and regularizing the other.

**Reaction Type Generalization Analysis**   Figure 2 reveals consistent performance across reaction types, with particular strength in complex transformations involving multiple bond changes. Traditional methods struggle with these due to template combinatorial explosion or invalid intermediate generation. Also is datasets is unbalanced across different class.

## 4.3 ABLATION STUDY

**Multi-task learning and domain specific adaptation's** contribution confirms our hypothesis that jointly learning related tasks creates a more robust and generalizable chemical representation as shown in Table 3. The bidirectional nature of our task set—incorporating both retrosynthesis and

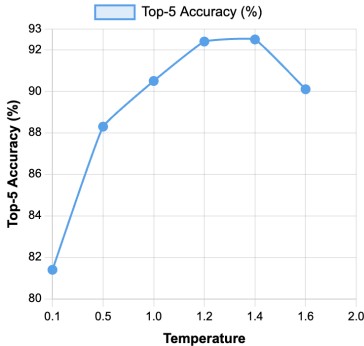

Figure 3: Top-5 accuracy v.s temperature.

Table 3: Ablation Study: Top-1 Accuracy (%) on USPTO-50k

| Method | Type Unknown | Type Known |
|---|---|---|
| SMILES with AAM | 74.8 | 75.4 |
| - Multi-task learning | 68.4 | 69.1 |
| - Task-spefic finetuning | 67.8 | 70.34 |
| - Plan token | 66.8 | 68.34 |
| SMILES with AAM-0 | 69.5 | 71.8 |
| - contrasting with AAM | 64.7 | 66.0 |
| Raw SMILES | 40.8 | 41.3 |

forward prediction—appears to create complementary learning signals that enhance model performance. At the same time, targeted adaptation to specific tasks further enhances performance. This two-stage approach balances generalization and specialization effectively.

**Plan-based reasoning** improve the performance when transformed from the foundation model to task-specific tuning. The planning mechanism not only improves accuracy but also provides interpretable reasoning traces for future training.

**Molecule representation and implicit contrastive learning** The AAM representation is most informative representation with high quality reaction information. However, the strong performance of AAM-0 (69.5%) indicates that our method remains highly effective even when mapping information is unavailable during inference, addressing a critical practical constraint. The raw SMILES baseline (40.8%) shows the importance of our representational innovations. Simply applying standard language modeling to SMILES strings is insufficient for high-accuracy reaction prediction.

## 4.4 EFFICIENCY AND PARAMETER SENSITIVITY

**Efficiency** Our compact 0.5B parameter model achieves superior performance while maintaining practical efficiency (24GB GPU memory). The robust performance across data conditions (mapped/unmapped, type-known/unknown) enhances real-world applicability where such information may be incomplete.

**Temperature sensitivity** Figure 3 presents a temperature sensitivity analysis. For top-1 accuracy, we employ greedy decoding (temperature = 0), which yields optimal performance for single-prediction scenarios. However, top-k accuracy improves with increasing temperature, reaching a maximum at T=1.6 before declining. Excessive randomness (T > 1.6) degrades performance.

## 5 LIMITATIONS AND FUTURE DIRECTIONS

While demonstrating strong performance and efficiency, we trained the base model over single step chemical reaction prediction. For complex multi-step reactions need further study. Though we show how good a compact a model can achieve, it is still valuable to train a larger model with more diverse data. Future work will focus on enhanced planning mechanisms for complex pathways and use the base model as agent for downstream tasks. Incorporation of practical constraints (feasibility, cost, safety) represents another important direction.

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

# A APPENDIX

## A.1 DATASET

The dataset statistics is shown in table 4.

Table 4: Dataset Information

| Dataset | Train Size | Validation Size | Test Size |
|---|---|---|---|
| USPTO_480k | 409,035 | 30,000 | 40,000 |
| USPTO_50k | 40,008 | 5,001 | 5,007 |
| USPTO_full | 810,496 | 101,311 | 101,311 |

## A.2 IMPLEMENTATION DETAILS

Our framework is implemented using the Qwen2.5-0.5B architecture as the foundation model. The Qwen2.5-0.5B model provides an optimal balance between computational efficiency and performance, featuring 0.5 billion parameters with a hidden dimension of 2,048, 16 attention heads, and 24 transformer layers. This architecture supports the extended sequence lengths required for chemical reaction prediction while maintaining manageable memory requirements.

Table 5: Model Architecture and Training Configuration

| Parameter | Value |
|---|---|
| **Base Model** | Qwen2.5-0.5B |
| Hidden dimension | 2,048 |
| Attention heads | 16 |
| Transformer layers | 24 |
| Maximum sequence length | 4,096 |
| **Optimization** | |
| Optimizer | AdamW |
| Learning rate | $1 \times 10^{-4}$ |
| $\beta_1, \beta_2$ | 0.9, 0.999 |
| Weight decay | 0.01 |
| Batch size | 32 |
| Warmup steps | 5,000 |
| Total training steps | 50,000 |
| **LoRA Configuration** | |
| Rank ($r$) | 64 |
| Alpha ($\alpha$) | 128 |
| Dropout | 0.1 |
| **Hardware** | |
| Device | A30 GPU |
| GPU Memory | 24GB |

We employ the AdamW optimizer with a learning rate of $1 \times 10^{-4}$, $\beta_1 = 0.9$, $\beta_2 = 0.999$, and weight decay of 0.01. Training utilizes a linear learning rate scheduler with warmup over 10% of the total training steps. The model processes sequences with a maximum length of 4,096 tokens to accommodate extended SMILES representations and plan-based reasoning steps.

For parameter-efficient fine-tuning, we implement Low-Rank Adaptation (LoRA) with rank $r = 64$ and scaling parameter $\alpha = 128$. LoRA adapters are applied to the query, key, value, and output projections in attention layers, as well as the gate, up, and down projections in feed-forward layers. This configuration enables efficient adaptation while adding only 0.2% additional parameters to the base model.

The complete training process requires approximately 72 hours on a single A30 GPU with 24GB memory. The combination of Qwen2.5-0.5B's compact architecture with LoRA fine-tuning ensures that our approach remains accessible while achieving state-of-the-art performance on chemical reaction prediction tasks.

## A.3 PROMPT FORMULATION

**Retrosynthesis Prompt**

Task: Retrosynthesis
Given the product SMILES: "product"
Predict the reactants required to synthesize this product.
### Instruction: - Think step-by-step to identify the reactants based on the product SMILES.
- Consider common retrosynthetic disconnections and reaction types (e.g., amide formation, esterification, nucleophilic substitution).
- Ensure the SMILES string is valid, includes atom mapping if present in the product, and uses '.' to separate multiple reactants.
- Note: This is an unmapped SMILES representation. If no atom mapping is provided or if atom mapping numbers are all 0 or -1, treat it as unmapped. Mapped and unmapped representations are similar but differ in format, with mapped including explicit atom correspondences. Still includes the atom mapping in the predicted reactants.
- Return the predicted reactants in SMILES format as a JSON object:
"reactants": "SMILES_string".

**Retrosynthesis with reaction type known Prompt**

Task: Retrosynthesis
Given the product SMILES: "product"and reaction class: "rxn_class
Predict the reactants required to synthesize this product.
### Instruction: - Think step-by-step to identify the reactants based on the product SMILES.
- Consider common retrosynthetic disconnections and reaction types (e.g., amide formation, esterification, nucleophilic substitution).
- Ensure the SMILES string is valid, includes atom mapping if present in the product, and uses '.' to separate multiple reactants.
- Note: This is an unmapped SMILES representation. If no atom mapping is provided or if atom mapping numbers are all 0 or -1, treat it as unmapped. Mapped and unmapped representations are similar but differ in format, with mapped including explicit atom correspondences. Still includes the atom mapping in the predicted reactants.
- Return the predicted reactants in SMILES format as a JSON object:
"reactants": "SMILES_string".

**Forward prediction Prompt**

Task: Forward prediction
Given the reactants SMILES: "reactants""
Predict the product of this reaction.
### Instruction:
- Think step-by-step to identify the product based on the reactants SMILES.
- Consider common reaction types (e.g., amide formation, esterification, nucleophilic substitution).
- Ensure the SMILES string is valid, includes atom mapping if present.
- Return the predicted product in SMILES format as a JSON object:
"product": "SMILES_string". """

## A.4 LLM USAGE

We use LLM when writing the paper for polishing and grammar checking.

