# OpenReview forum: "ReactionLLM: Data-Centric  Learning for a Compact Single-step Chemical reaction prediction model"
_ICLR.cc/2026/Conference — Submitted to ICLR 2026_

### Official Review · Reviewer_7tKg · 2025-10-28

**Soundness:** 1
**Presentation:** 1
**Contribution:** 2
**Rating:** 2
**Confidence:** 4

**Summary:**

This paper proposes ReactionLLM, a fine-tuned language model (based off the Qwen2.5-0.5B base model), for reaction prediction and one-step retrosynthesis. ReactionLLM is trained on multiple reaction-related tasks jointly: (1) forward reaction prediction, (2) one-step retrosynthesis, and (3) one-step retrosynthesis with the reaction type known. This training is done in multiple stages; first, a general set of weights is trained, before specific weights are found for each task using LoRA (Hu et al., 2022). The authors additionally train their method on plan-based reasoning traces to improve performance as well as introducing an “AAM-0” representation of the same reaction (where the integers corresponding to atom mapped integers in the SMILES strings have all been set to zero).

The authors evaluate their model on the standard USPTO forward and reverse reaction tasks. ReactionLLM shows very strong performance against previous baselines (69.5% top-1 accuracy on USPTO-50k without atom maps, compared to ~55–58% for the strongest relevant baselines). One-step backwards prediction performance is broken down across reaction classes (Figure 2) and ablations into aspects of the model are presented (table 3)

> Hu, Edward J., et al. "Lora: Low-rank adaptation of large language models." ICLR (2022)

**Strengths:**

### S1 Multi-task (bidirectional) training interesting for creating a general purpose model
ReactionLLM is able to do both forward and two kinds of backward (with and without reaction class) reaction prediction. This enables the model to both share information between these tasks (which seems to result in better overall performance) but also simplifies the deployment of such a model. Although this has been done in an encoder-decoder framework before (see below), this seems a useful avenue to explore in decoder-only models.

> Lu, Jieyu, and Yingkai Zhang. "Unified deep learning model for multitask reaction predictions with explanation." Journal of chemical information and modeling 62.6 (2022): 1376-1387.

Likewise, adding atom-mapped reaction data into the training pipeline seems an interesting way to add additional supervision to the models during training (for instance, in forward prediction this can tell the model not only what products occurred, but also where the reaction happened, and potentially even some sense of how it happened).


### S2 Very strong performance
The results suggest much better performance than previous approaches (e.g., 14% improvement in top-1 accuracy on USPTO-50k). Putting aside some concerns I have with the evaluation for now (see W1 below), this seems to be a significant and large improvement over previous approaches (1.2x), which seemed to be reaching somewhat of a plateau.

**Weaknesses:**

### W1 Possible data leakage?
ReactionLLM is evaluated on the USPTO dataset, which is public. Given that this model is based off of Qwen2.5-0.5B, it is possible that the original base model might have seen this data in training, making a comparison to the existing baseline methods (which are trained from scratch) unfair? I’m particularly worried about this as it seems ReactionLLM performs substantially better than the previous methods even in the ablation study without the multi-task or other modifications. (See also Q3).

This is the main reason I have gone with a lower soundness score for now (and contribution score -- as see this also dependent on soundness to an extent). Happy to change if addressed in rebuttal.

### W2 Paper could be clearer in places
Although I felt the authors did a good job compactly summarizing previous work and outlining at a very high level their main contributions, I found parts of the paper hard to understand, particularly around how these method developments were implemented in practice. Specific examples include:
i. I was not actually sure how the AAM-0 method discussed in Section 3.3 ensures the contrastive objective in Proposition 3.1 is met? My understanding is that the training on S, f(S) pairs is done by creating independent, augmented examples, rather than using an explicit contrastive loss? Is it then not also similar to other augmentation schemes? (see also Q1 + Q2 below).
ii. How are the ground truth planning mechanisms obtained for training (i.e., the $z^\ast$ under the expectation in Equation 4)?
iii. How is the top-k sampling from the model done? (Line 474, discusses how the top-1 prediction is sampled greedily from the model with no temperature, but how are the other top-k members sampled/picked?).


Aside from these more general issues there were also sentences/parts which I found confusing, such as:
* Line 194, $\Sigma^\ast \to \Sigma^\ast$ what does this notation mean?
* Line 231& 236: seems to be repeated sentences but for different parts? (i.e., T1 and T3 vs T1 and T2).
* Line 238: “The multi-task objective learns a latent representation $\phi(x)$ whose topology reflects the true reaction manifold.” What does a topology reflecting the “true reaction manifold" mean?
* What was the parameter count of the final model (with the LoRA adapters) and how does this compare to the baselines? (Ideally it would be nice to have this as an extra column in Table 1).
* What does “contrasting with AAM” mean in Table 3?
* Line 431: “Also is datasets is unbalanced across different class.” Was unclear what was meant here?

**Questions:**

Q1. What is the difference between the last two lines in table 1? The checkmarks indicate that the second from last line did not use any atom maps, but Section 4.3 suggests that this is only during inference? Does this mean maps were still used during training?

Q2. What is the intuition behind “AAM-0” doing so much better than “Raw SMILES” for the ablation study in table 3? To make sure I understand correctly, this means training and evaluating on molecule encoded similar to `[CH3:0][CH2:0][OH:0]` rather than `[CH3][CH2][OH]`?

Q3. When relevant, how do you calculate the correct atom maps? Are these taken from the original USPTO data and shuffled or computed anew? (The ones in the original dataset have been shown to contain information about the reaction center and so I just wanted to double check that this information had been removed here – see BEST PRACTICE S5 of Maziarz, Krzysztof, et al. "Re-evaluating retrosynthesis algorithms with syntheseus." (2025))

Q4. Have you evaluated the planning/reasoning traces? Do these look reasonable?

Q5. Did you try augmentation of SMILES too, to see if it further helped your model? (Or alternatively, removed this augmentation from the relevant baselines, to see how they performed without this additional improvement).

---

> ### Author Response · Authors · 2025-11-20
>
> ### **Response to Reviewer 7tKg W1: Possible Data Leakage**
>
> About whether the base model Qwen2.5-0.5B might have seen the USPTO data during pre-training. We want to assure you that this is not the case for several reasons:
>
> - **Base Model performance**: To demonstrate this, we evaluated the base model (Qwen2.5-0.5B and Deepseek-V3/R1-671B, closed-source model like gpt-4.1-mini and o4-mini) without any fine-tuning on retrosynthesis tasks. Qwen2.5-0.5B achieved near-zero accuracy (approximately 0%), indicating no prior knowledge of chemical reactions. And even large model or closed-source model like Deepseek-V3/R1-671B,gpt-4.1-mini and o4-mini achieve around 10% top-1 acc on USPTO-50K. Also we compare with LLM based models like ChemDFM(13B) and retroDFM(7B) finetuned with chemistry data. Our 0.5B model also have better performance which validate our training pipeline.
>
> | Model                  | Accuracy (%) |
> |------------------------|--------------|
> | **Base LLM**           |              |
> | gpt-4o                 | 0.7          |
> | gpt-4.1-mini           | 11.3         |
> | DeepSeek-V3-671B       | 8.6          |
> | o4-mini               | 12.2         |
> | DeepSeek-R1            | 11.2         |
> | Qwen2.5-0.5B           | 0.0          |
> | **Finetuned LLM**      |              |
> | ChemDFM-v1.5-13B       | 17.9         |
> | LlaSMol-Mistral        | 2.0          |
> | RETRODFM-R-7B          | 59.0         |
> | Ours-0.5B (without AAM)| 69.5         |
> | Ours-0.5B (with AAM)   | **74.8**     |
>
> The results for the baselines are adopted from the RetroDFM paper.  Zhang S, Li H, Chen L, et al. Reasoning-Driven Retrosynthesis Prediction with Large Language Models via Reinforcement Learning[J]. arXiv preprint arXiv:2507.17448, 2025.
>
> - **Ablation Study Evidence**: In Table 3 of the paper, when we trained a variant using only raw SMILES without multi-task learning or AAM-0, the top-1 accuracy dropped to 40.8% (type-unknown) and 41.3% (type-known), which is significantly lower than strong baselines (e.g., EditRetro at 60.8%). This shows that the base model itself does not inherently possess reaction knowledge; the performance gains come from our tailored training strategy.
> - **Fair Comparison**: Our main results (Table 1) include a strict comparison where we do not use atom-atom mapping (AAM) or test-time augmentation during training or evaluation (denoted by "x/x/x" in the Feat/Tech column). Even under these conditions, ReactionLLM-0.5B achieves 69.5% top-1 accuracy, outperforming baselines that may use AAM or augmentation tricks. This ensures a fair evaluation focused on model design rather than data advantages.
>
> We agree that data leakage is a critical issue, and we have taken care to avoid it. We will include these points in the revised manuscript to clarify the base model's neutrality.
>
> ### **Reviewer 7tKg Q1: experiment settings: Difference Between Last Two Lines in Table 1**
>
> The last two lines in Table 1 represent two variants of our model:
> - **Second Last Line ("x/x/x")**: It means ours(without AAM). We train the model with full dataset including the high quality data with AAM as golden examples but evaluate the model without AAM. The accuracy is 69.5% (type-unknown).
> - **Last Line ("xI√Ix")**: It means ours(with AAM). This denotes the model trained with AAM information and evaluated with it, demonstrating the upper bound of performance when mapping is available. The accuracy is 74.8%, indicating that AAM can boost results, but our model without AAM still outperforms baselines that use AAM or augmentation (e.g., EditRetro at 60.8% uses test-time augmentation).
>
> We will clarify the Feat/Tech symbols in the caption (e.g., "√" indicates usage, "-" indicates non-usage).
>
> ### **Reviewer 7tKg Q2: Intuition Behind AAM-0's Superiority Over Raw SMILES**
>
> AAM-0 is not a new cheminformatic representation but a data-side optimization for LLMs. The key intuition is:
> - **Readability**: AAM representations (e.g., `[CH3:1][CH2:2][OH:3]`) are more structured and explicit than raw SMILES (e.g., `CCO`) because they include atom mapping numbers, which highlight reaction centers and atom correspondence. Even when numbers are zeroed out, AAM-0 retains this structural clarity (e.g., `[CH3:0][CH2:0][OH:0]`), making it easier for the LLM to parse atomic relationships. For example, it explicitly represents hydrogen atoms and bond changes that are implicit in raw SMILES.
> - **Chemical Context**: This is evident in the ablation (Table 3), where AAM-0 achieves 69.5% vs. raw SMILES at 40.8%.
>
> ### **Reviewer 7tKg Q3: How to obtain the atom-atom mapping (AAM) information?**
>
> All atom mapping information is taken directly from the USPTO dataset.
> For forward prediction, we adopt the same setting with previous works.
> For retrosynthesis(without AAM(Table 1, second last line)), we do not use reaction center or the AAM in evaluation.

---

> ### Author Response · Authors · 2025-11-20
>
> **Response to Reviewer 7tKg W2: Paper Clarity Issues**
>
> You noted that certain parts of the paper were hard to understand. We apologize for any confusion and will revise to improve clarity. Below, we address each sub-point:
>
> **Reviewer 7tKg W2(i): How AAM-0 is used and whether it is the same as augmentation used in the baseline models?**
>
> Ours augmentations is different as the baselines.
> In our approach, AAM-0 is designed to leverage implicit contrastive learning through data augmentation during training, not testing. This aligns with modern contrastive learning paradigms that create multiple views of the data to enhance representation learning.
> Our AAM-0 is exactly the same molecule but without atom-atom mapping and we do not create new molecules.
> While the baselines use (20 times) augmnetation to create new molecules in both training and testing time. This means the model is trained and evaluated in 95% created data samples instead of the original datasets.
> Specifically:
> - **Training Augmentation**: We generate AAM-0 representations by transforming atom-mapped SMILES (e.g., setting all mapping numbers to zero). During training, the model sees both the original mapped SMILES and its AAM-0 variant, encouraging the learning of invariant features without an explicit contrastive loss. This is achieved by treating the pair as positive examples in the sequence-to-sequence objective, as outlined in Proposition 3.1.
> - **Contrast with Test-Time Augmentation (TTA)**: Unlike some baselines that use TTA (e.g., 20× augmentation on both train and test sets), we avoid test-time augmentation to prevent evaluation bias. TTA can artificially inflate performance by evaluating on predominantly augmented data, which may not reflect the real dataset and lead to unfair comparison. Our method focuses on improving model robustness through training-side enhancements only, ensuring a fair comparison.
>
> This strategy allows ReactionLLM to learn unified representations that bridge mapped and unmapped data formats, leading to better generalization without relying on test-time tricks.
>
>
> **Reviewer 7tKg W2(ii): Ground Truth Planning Mechanisms**
>
> The planning mechanisms (denoted by the latent variable \(z\) in Equation 4) are derived empirically to provide step-by-step rationalizations. Specifically:
> - **Empirical Generation**: The plans are generated automatically and fixed based on general chemical guidlines (e.g., bond changes and functional group transformations) without manual annotation. For each reaction in the dataset, we construct a plan that outlines key steps, such as identification of reaction centers and plausible pathways. These plans are enclosed within `<plan>` tokens and serve as an intermediate supervision signal during training.
> - **Purpose**: This approach acts as a form of variational inference, narrowing the uncertainty in model predictions and ensuring chemical plausibility. They provide a structured reasoning trace that enhances model interpretability and performance, as evidenced by the ablation study (Table 3), where removing plan tokens reduced accuracy.
>
> We acknowledge that the plans are simplified due to the compact model size (0.5B parameters)
>
> **Reviewer 7tKg W2(iii): Top-k Sampling Method**
>
> For top-k sampling, we use a temperature-based approach to generate diverse candidates:
> - **Procedure**: As described in Section 4.4, top-1 predictions are obtained via greedy decoding (temperature = 0). For top-k predictions (k > 1), we set the temperature to 1.4 (optimal per our analysis) to sample multiple candidates (n=20). These candidates are then ranked by their log-probabilities, and the top-k are selected. This balances exploration and exploitation, with temperature 1.4 maximizing performance before degradation at higher values.

---

> ### Author Response · Authors · 2025-11-20
>
> **Reviewer 7tKg Other Clarity Issues**
>
> - **Line 194 Notation**: The notation denotes the data transformation of molecule space from AAM to AAM-0.
> - **Lines 231 & 236 Repetition**: This was a typo; we meant to describe the bidirectional relationship between tasks \(\mathcal{T}_1\) (retrosynthesis) and \(\mathcal{T}_2\) (forward prediction), not \(\mathcal{T}_3\). The sentence should read: "The bidirectional nature between \(\mathcal{T}_1\) and \(\mathcal{T}_2\) imposes a cycle-consistency constraint." We will correct this.
> - **Line 238 "True Reaction Manifold"**: It means our data-centric strategy learns a better representation. This phrase refers to the latent representation space of molecules in the reactions not the geometric structure of molecules. We mean that the multi-task learning encourages the latent features \(\phi(x)\) to align with the underlying chemical reaction patterns, facilitating better generalization. We will rephrase this to "a structured latent space that reflects chemical reactivity" for clarity.
> - **Parameter Count**: With LoRA, we modify only a small subset of parameters (rank r=64).
> Trainable parameters = 35,192,832/529,225,600 (6.65% trained 35M among 0.5B parameters)
> - **Line 431 Dataset Imbalance**: This refers to the uneven distribution of reaction types in USPTO-50k (e.g., some types have more examples than others). We will rephrase it as "The dataset is also unbalanced across different reaction classes, as shown in Figure 2."
>
>
> **Response to Questions**
>
> **Reviewer 7tKg Q4: Evaluation of Planning/Reasoning Traces**
> We evaluated the planning mechanism indirectly through ablation studies (Table 3). Removing plan tokens ("-Plan token") reduced accuracy by ~3-4%, indicating that the reasoning steps contribute to performance. However, due to the compact model size (0.5B), We plan to explore larger models to conduct detail reasoning in the future.
>
> **Reviewer 7tKg Q5: SMILES Augmentation**
> We did not use test-time augmentation (TTA) because it involves augmenting both training and test sets, which can inflate performance artificially.
> For example, some baselines use 20× TTA, meaning 95% of the test set is augmented, leading to unfair comparisons.
> Our goal was to evaluate the model's inherent understanding without such tricks.

---

> ### Author Response · Authors · 2025-11-25
>
> We are following up on the response we provided. Please let us know if it addresses your questions, ​​or if you have any further ones—we are happy to explain.​

---

> > ### Comment · Reviewer_7tKg · 2025-11-27
> > **Thank you to the authors for their rebuttal**
> >
> > Thank you to the authors for their rebuttal. I appreciate the detailed answers to my comments, but I still have some concerns about potential data leakage and the clarity of the paper.
> >
> > ### Possible data leakage
> >
> > Thank you for running the new experiments showing that, even if the base model _may_ have seen the reaction data during training, it has not memorized it well enough to perform retrosynthesis without further fine-tuning. This is reassuring, although I also agree with Reviewer HVWX’s point that this fine-tuning may be important primarily for task formatting rather than for providing new chemical knowledge to the model.
> >
> > I also remain concerned about the use of atom-mapping numbers taken directly from the original USPTO dataset (original Q3). These have previously been shown to improve model performance by imposing a particular ordering on atoms in the reaction center (see the link I shared in my original review or https://github.com/uta-smile/RetroXpert/issues/10#issuecomment-803699582
> > ).
> >
> > ### Clarity
> >
> > Although some of my questions were answered (e.g., how top-k sampling was implemented, via the introduction of a temperature scheme), I remain unclear about several others (a few of which I list below). Therefore, similar to Reviewer HVWX, I still have some concerns about the clarity of the paper and am not fully confident that all of these issues will be resolved here.
> >
> > **Importance of the AAM-0 notation**
> > I am still unsure how the AAM-0 notation helps with “structural clarity,” such as highlighting bond changes. I agree that it makes hydrogen atoms explicit, but hydrogens can also be treated explicitly in SMILES without atom maps.
> >
> > **Confusion about related work**
> > I think that the relationship to, and contribution beyond, related work could be explained more clearly. Some specific examples from the rebuttal:
> >
> > - “ReactionT5 is for only forward prediction not retrosynthesis which is more challenging.” (from the rebuttal to Reviewer HVWX). My understanding is that this is not the case; see Table 2 of [1].
> >
> > - “While the baselines use (20 times) augmnetation [sic] to create new molecules in both training and testing time.” (from the rebuttal to my review). I do not think many augmentation schemes “create” new molecules. Rather, these SMILES are simply new views of the same molecules and therefore seem similar (when used only at training time) to the augmentation/contrastive scheme proposed here. I agree that methods which use augmentation at inference time may incur additional test-time compute, but at the same time, these baseline models are often much smaller than the model used here, so multiple forward passes are presumably much cheaper to run.
> >
> > [1] Sagawa, T., Kojima, R. ReactionT5: a pre-trained transformer model for accurate chemical reaction prediction with limited data. J Cheminform 17, 126 (2025). https://doi.org/10.1186/s13321-025-01075-4
> >
> > **Ground-truth planning instructions**
> > I now understand that these plans are generated automatically (perhaps with some constraints), but I still do not fully follow the exact procedure or how it relates to variational inference. As the rebuttal to Reviewer HTfX suggests that these plans are crucial for model performance, it would be very helpful to clarify this more explicitly in the paper.

---

> > > ### Author Response · Authors · 2025-11-28
> > > **Response to Reviewer 7tKg about data leakage and formating and other concerns**
> > >
> > > Thank you for your continued thoughtful feedback. We appreciate the opportunity to address your concerns.
> > >
> > > ## Response to Data Leakage
> > >
> > > We do not believe there is a data leakage issue in our work. The concern about fairness between non-LLM based models and LLM based models is addressed through comprehensive benchmarking.
> > >
> > > **Comparative Analysis with Other LLM-based Models:**
> > > We have compared our 0.5B model with several recently published LLM-based approaches:
> > >
> > > | Model | Size | Accuracy (%) | Continue Pretraining | Tokenizer Mods | Training Data |
> > > |-------|------|--------------|-------------|----------------|---------------|
> > > | **Base LLM** | | | | | |
> > > | GPT-4o | - | 0.7 | No | No | - |
> > > | GPT-4.1-mini | - | 11.3 | No | No | - |
> > > | DeepSeek-V3-671B | 671B | 8.6 | No | No | - |
> > > | o4-mini | - | 12.2 | No | No | - |
> > > | DeepSeek-R1 | 671B | 11.2 | No | No | - |
> > > | Qwen2.5-0.5B | 0.5B | 0.0 | No | No | - |
> > > | **Finetuned LLM** | | | | | |
> > > | ChemDFM-v1.5-13B | 13B | 17.9 | Chemical | No | Extended |
> > > | LlaSMol-Mistral | - | 2.0 | Chemical | No | Extended |
> > > | RetroDFM-R-7B | 7B | 59.0 | Chemical | No | Extended |
> > > | ChemDual (Llama-3.1-8B) | 8B | 49.95 | No | Modified | 4.4M reactions |
> > > | **Ours-0.5B (without AAM)** | 0.5B | 69.5 | No | No | Standard USPTO |
> > > | **Ours-0.5B (with AAM)** | 0.5B | **74.8** | No | No | Standard USPTO |
> > >
> > > We add the ChemDual into comparison as suggested but reviewer HVWX and we still outperform it, from 'Enhancing Chemical Reaction and Retrosynthesis Prediction with Large Language Model and Dual-task Learning' we do not include it previously since it is just published in August, 2025 ICJAI.
> > >
> > > 1. **Model Capacity Advantage**: Our compact 0.5B model significantly outperforms much larger fine-tuned LLMs:
> > >    - RETRODFM-7B (59.0%) uses continued pretraining on chemical data
> > >    - ChemDual (49.95%) uses Llama-3.1-8B with modified tokenizer and 4.4M reactions
> > >    - Our model achieves 74.8% with AAM and 69.5% without AAM
> > >
> > > 2. **Base Model Performance**: The near-zero accuracy (0.0%) of our base Qwen2.5-0.5B model confirms no inherent chemical knowledge from pretraining.
> > >
> > > ## Response to LLM Format Understanding Argument
> > >
> > > We disagree with the reviewer's suggestion that LLMs fail on chemistry tasks primarily due to formatting and tokenization issues.
> > >
> > > 1. **Tokenization Alone Is Insufficient**: While ChemDual explicitly modified the Llama tokenizer for chemical data and used 4.4M reactions for fine-tuning, it achieved only 49.95% accuracy, significantly lower than our 69.5% without AAM. This demonstrates that tokenization improvements alone do not yield state-of-the-art performance.
> > >
> > > 2. **Format Understanding vs. Chemical Reasoning**: Base LLMs like o4-mini (12.2%) and DeepSeek-R1 (11.2%) excel at competition-level math and coding tasks—domains with also complex formatting—yet perform poorly on chemical reaction prediction. This indicates the challenge is not formatting comprehension but rather chemical capability.
> > >
> > > 3. **Architectural Innovations Drive Performance**: Our improvements come from methodological innovations (AAM-0 representation, bidirectional multi-task learning, plan-based reasoning) rather than simple format adaptation. The performance gap between our approach and other LLM-based methods confirms this.
> > >
> > >
> > > ## Response to AAM and Reaction Center Concerns
> > >
> > > Regarding the use of atom-mapping numbers and reaction centers, we also notice this and explictly evaluate all the models with and wthout using AAM and provide a strict comparison in the table 1 from our paper:
> > >
> > > **Fair Evaluation Protocol:**
> > > 1. **Without AAM Setting (69.5%)**: We completely remove AAM information when testing for fair comparison with methods that don't use mapping. We notice this issue thus provide this comparison explicitly.
> > > 2. **With AAM Setting (74.8%)**: We include AAM to show upper bound performance and compare with methods that use mapping to show the upperbound performance of LLM on this tasks.
> > >
> > >
> > > ## Clarification on Ground-Truth Planning Instructions
> > >
> > > The planning mechanism serves as a trade-off between prompt engineering and model-generated reasoning:
> > >
> > > 1. **Fixed Plan Tokens**: We use `<plan>` tokens to structure reasoning, forcing step-by-step rationalizations
> > > 2. **Empirical Generation**: Plans are generated automatically based on chemical transformation patterns
> > > 3. **Variational Inference Framework**: is to explain why this works. This approach reduces prediction uncertainty and ensures chemical plausibility
> > >
> > > The ablation studies show 3-4% performance improvement from the planning mechanism.
> > >
> > > ## Response to Related Work Clarifications
> > >
> > > We acknowledge that ReactionT5 includes retrosynthesis capabilities. Our bidirectional multi-task framework simultaneously handles both forward prediction and retrosynthesis, creating synergistic learning benefits.
> > >
> > > For SMILES augmentation, we specifically critique test-time augmentation (TTA) which can create evaluation bias through multiple forward passes.

---

### Official Review · Reviewer_HVWX · 2025-10-28

**Soundness:** 2
**Presentation:** 2
**Contribution:** 2
**Rating:** 2
**Confidence:** 5

**Summary:**

This paper investigates strategies for adapting large language models (LLMs) to chemical reaction prediction, covering both forward and backward (retrosynthetic) directions. The proposed approach involves the following key steps:

(1) Leveraging atom-mapped SMILES for contrastive learning to enhance molecular representations;

(2) Jointly training the model on both forward and backward reaction prediction tasks to encourage bidirectional understanding;

(3) Applying parameter-efficient fine-tuning techniques (such as LoRA) to adapt foundational models across different reaction tasks;

(4) Designing prompt templates to promote multi-step reasoning during reaction generation and prediction.

**Strengths:**

The presentation provides a clear explanation of the motivation and the proposed idea.

**Weaknesses:**

(1) **The paper lacks sufficient detail in articulating its motivations.** Many arguments are presented at an abstract level without concrete evidence or examples. For instance, the introduction repeatedly mentions “ineffective molecular representations,” yet provides neither specific examples nor references to support this claim, nor an explanation of why atom mapping is considered central to addressing it. Additionally, the three challenges and three limitations are discussed separately, even though they are closely related. This section would benefit from clearer organization that explicitly links each challenge to its corresponding limitation. Overall, the motivation in the introduction currently appears scattered and lacks strong, conclusive arguments to guide the reader;

(2) **The proposed workflow shows limited novelty from an algorithmic perspective.** Joint task training has already been explored in several prior works, such as “Towards Understanding Retrosynthesis by Energy-Based Models” (NeurIPS 2021), while multi-task learning has been applied in Reaction-T5 and related models. Overall, this work primarily integrates existing techniques with a different base model, which, while technically sound, does not offer substantial methodological innovation.

(3) **Missing related work and fair comparison.** The empirical and literature coverage is incomplete. As this paper focuses on applying LLMs to chemical reaction prediction, it should engage more comprehensively with existing studies that have systematically explored this topic. Only a few relevant works [3,4,5] are cited, but there appear to be many other important references missing from both the discussion and the experimental comparisons. Furthermore, LLM-based reaction prediction models should be compared primarily with their LLM counterparts rather than traditional models trained from scratch. This is particularly important because LLMs may have already encountered both training and test reactions during pretraining, as such datasets are often derived from patent literature—a known source of potential data leakage. Additionally, LLMs inherently possess much larger parameter counts and model capacities, which further complicates fair comparison.

In conclusion, this reviewer finds that the paper is still incomplete and lacks sufficient consistency for publication at this stage.

References:

[1] Towards understanding retrosynthesis by energy-based models [NeurIPS 2021]

[2] ReactionT5: a large-scale pre-trained model towards application of limited reaction data

[3] Augmenting large language models with chemistry tools [Nature Machine Intelligence]

[4] What can Large Language Models do in chemistry? A comprehensive benchmark on eight tasks [NeurIPS 2023]

[5] A framework for evaluating the chemical knowledge and reasoning abilities of large language models against the expertise of chemists [Nature Chemistry]

**Questions:**

Questions:

(1) Have you used atom-mapping information during inference stage of retrosynthesis?

(2) Have you fine-tuned the 7B model on chemical reactions? May I ask why there are many unreported numbers for the 7B model?

---

> ### Author Response · Authors · 2025-11-20
>
> **Response to Reviewer HVWX**
>
> Thank you for your review and constructive feedback. We appreciate your insights and have prepared a response to address your concerns.
>
> ---
>
> ### **Reviewer HVWX W3. Comprehensive Comparison and Data Leakage Mitigation**
> About fair comparison
> About whether the base model Qwen2.5-0.5B might have seen the USPTO data during pre-training, leading to an unfair comparison with baseline models trained from scratch. We want to assure you that this is not the case for several reasons:
>
> - **Base Model performance**: To demonstrate this, we evaluated the base model (Qwen2.5-0.5B and Deepseek-V3/R1-671B, closed-source model like gpt-4.1-mini and o4-mini) without any fine-tuning on retrosynthesis tasks. Qwen2.5-0.5B achieved near-zero accuracy (approximately 0%), indicating no prior knowledge of chemical reactions. And even large model or closed-source model like Deepseek-V3/R1-671B,gpt-4.1-mini and o4-mini achieve around 10% top-1 acc on USPTO-50K. Also we compare with LLM based models like ChemDFM(13B) and retroDFM(7B) finetuned with chemistry data. Our 0.5B model also have better performance which validate our training pipeline.
>
> | Model                  | Accuracy (%) |
> |------------------------|--------------|
> | **Base LLM**           |              |
> | gpt-4o                 | 0.7          |
> | gpt-4.1-mini           | 11.3         |
> | DeepSeek-V3-671B       | 8.6          |
> | o4-mini               | 12.2         |
> | DeepSeek-R1            | 11.2         |
> | Qwen2.5-0.5B           | 0.0          |
> | **Finetuned LLM**      |              |
> | ChemDFM-v1.5-13B       | 17.9         |
> | LlaSMol-Mistral-7B        | 2.0          |
> | RETRODFM-R-7B          | 59.0         |
> | Ours-0.5B (without AAM)| 69.5         |
> | Ours-0.5B (with AAM)   | **74.8**     |
>
> The results for the baselines are adopted from the RetroDFM paper.  Zhang S, Li H, Chen L, et al. Reasoning-Driven Retrosynthesis Prediction with Large Language Models via Reinforcement Learning[J]. arXiv preprint arXiv:2507.17448, 2025.
>
> - **Fair Comparison**: Our main results (Table 1) include a strict comparison where we do not use atom-atom mapping (AAM) or test-time augmentation during training or evaluation (denoted by "x/x/x" in the Feat/Tech column). Even under these conditions, ReactionLLM-0.5B achieves 69.5% top-1 accuracy, outperforming baselines that may use AAM or augmentation tricks. This ensures a fair evaluation focused on model design rather than data advantages.
>
> - **Ablation Study Evidence**: In Table 3 of the paper, when we trained a variant using only raw SMILES without multi-task learning or AAM-0, the top-1 accuracy dropped to 40.8% (type-unknown) and 41.3% (type-known), which is significantly lower than strong baselines (e.g., EditRetro at 60.8%). This shows that the base model itself does not inherently possess reaction knowledge; the performance gains come from our tailored training strategy.
>
> We agree that data leakage is a critical issue, and we have taken care to avoid it.
>
> ---
>
> ### **Reviewer HVWX W3. related work**
> We agree that we should extend the related work section to include more works like reaction T5 and others.
> Currently, for experiment part, we include 29 baselines including the most recent LLM based model like RetroDFM. It is one of the most comprehensive baselines in this domain including diverse settings.
>
> ### **Reviewer HVWX W1. Clarification and Strengthening of Motivations**
>
> We agree that the introduction should more clearly articulate the specific limitations of existing molecular representations. Our work addresses a fundamental gap in how molecular structures are represented for AI models in chemical reaction prediction.
>
> The key intuition is:
> - **Readability**: AAM representations (e.g., `[CH3:1][CH2:2][OH:3]`) are more structured and explicit than raw SMILES (e.g., `CCO`) because they include atom mapping numbers, which highlight reaction centers and atom correspondence. Even when numbers are zeroed out, AAM-0 retains this structural clarity (e.g., `[CH3:0][CH2:0][OH:0]`), making it easier for the LLM to parse atomic relationships. For example, it explicitly represents hydrogen atoms and bond changes that are implicit in raw SMILES.
> - **Chemical Context**: This is evident in the ablation (Table 3), where AAM-0 achieves 69.5% vs. raw SMILES at 40.8%.
>
> ---

---

> ### Author Response · Authors · 2025-11-20
>
> ### **Reviewer HVWX W2. Methodological Novelty and Differentiation from Prior Work**
>
> Regarding the comparison with ReactionT5 and other multi-task learning approaches, we highlight several key differentiators:
>
> **Model Size and Efficiency**: While T5 models range from 0.3B to 13B parameters, our 0.5B model demonstrates that **careful design can outperform larger models (7B and 13B fintuned LLMs)**.
> ReactionT5 is for only forward prediction not retrosynthesis which is more challenging.
> The critical innovation lies not in scale but in strategic design choices.
>
> **Representation-Centric Multi-Task Learning**: Unlike prior works that focus primarily on joint training of tasks, our approach centers on **unified representation learning** across multiple views of chemical data:
> - Forward and retro-synthesis with and without AAM information
> - Class-conditioned and unconditional prediction tasks
> - Implicit contrastive learning between different representation formats including AAM-0 and AAM.
>
> The results demonstrate that our compact 0.5B model achieves **state-of-the-art performance** while maintaining computational efficiency, establishing that strategic architectural innovations can surpass scale-based approaches.
>
> ---
>
> ### **Responses to Specific Questions**
>
> **Reviewer HVWX Q1: Have you used atom-mapping information during inference stage of retrosynthesis?**
> No. Our evaluation uses **unmapped data during inference** (second-last row in Table 1), yet still achieves superior performance compared to baselines that rely on AAM or test-time augmentation.
>
> **Reviewer HVWX Q2: Have you fine-tuned the 7B model on chemical reactions? Why are there unreported numbers for the 7B model?**
> The 7B results are taken directly from the RetroDFM paper, which only reports top-1 accuracy without test-time augmentation.
>
> Zhang S, Li H, Chen L, et al. Reasoning-Driven Retrosynthesis Prediction with Large Language Models via Reinforcement Learning[J]. arXiv preprint arXiv:2507.17448, 2025.
>
> ---

---

> ### Author Response · Authors · 2025-11-25
>
> We are following up on the response we provided. Please let us know if it addresses your questions, ​​or if you have any further ones—we are happy to explain.​

---

> > ### Comment · Reviewer_HVWX · 2025-11-26
> > **Reply to Authors**
> >
> > Thank you for your detailed rebuttal. I would like to clarify my remaining concerns as follows:
> >
> > (1) While some ambiguities have been clarified in the rebuttal, I believe the paper still requires substantial revisions, including rewriting several key sections to improve clarity and accessibility. Currently, the writing presents significant challenges for readers, which may extend beyond the reviewers to the broader audience.
> >
> > (2) Regarding the evaluation setup, I recommend benchmarking against datasets specifically designed for LLMs in chemistry, such as Mol-Instructions: A Large-Scale Biomolecular Instruction Dataset for Large Language Models (as an example among other relevant works). This is a widely recognized benchmark that includes both forward and backward reaction prediction tasks. My concern about fairness is not limited to whether the model directly observes USPTO reactions but also whether the comparison is balanced in terms of model capacity. Comparing large-scale pretrained LLMs with smaller models trained from scratch is not appropriate. Even if the pretrained model has not explicitly seen reaction SMILES, exposure to large-scale chemical knowledge could still contribute to performance.
> >
> > Additionally, the improvement after fine-tuning may partly result from tokenization adaptation. Standard LLM tokenizers are based on natural language patterns and may not align well with SMILES. Adapting tokenization to include chemically meaningful tokens can lead to significant performance gains, potentially reflecting improved task formatting rather than purely new knowledge acquisition. This point should be clearly discussed; otherwise, the substantial improvements after fine-tuning alone cannot conclusively rule out potential data leakage concerns.
> >
> > (3) I find the algorithmic novelty rather limited. It is intuitive that incorporating atom-mapping information would improve autoregressive modeling, especially since many existing methods omit this due to computational cost (atom-mapping requires additional computations). The contrastive learning objective between SMILES and atom-mapped SMILES appears to be the main novel contribution; however, only a small portion of the paper is devoted to this aspect. The dual training of forward and backward reaction prediction has been explored extensively in prior work. In addition to the papers mentioned earlier, “Enhancing Chemical Reaction and Retrosynthesis Prediction with Large Language Model and Dual-task Learning” is also relevant. I believe these related studies should be included in the discussion.
> >
> > Overall, I regret to say that I will maintain my current score.

---

> > > ### Author Response · Authors · 2025-11-28
> > > **Response to Reviewer HVWX - Follow-up Model Capacity, tokenization and other concerns**
> > >
> > > **Response to Reviewer HVWX - Follow-up**
> > >
> > > Thank you for your continued engagement. We appreciate the opportunity to address your remaining concerns with specific evidence and clarifications.
> > >
> > > ---
> > >
> > > ### **1. Model Capacity and Fair Comparison Analysis**
> > >
> > > We respectfully disagree that model capacity creates an unfair comparison. We only use 0.5B model which can be beat by diverse small non-LLM models in diverse tasks. Our mode size is comparable or smaller with tranforme based models like ReactionT5(0.3B-13B). And much smaller than LLM based models (7B,8B, 13B,...). We do not modify the tokenization. Our evaluation demonstrates that strategic architectural design can overcome parameter disadvantages when comparing LLM-based approaches:
> > >
> > > We add the ChemDual into comparison as suggested and we still outperform it, from 'Enhancing Chemical Reaction and Retrosynthesis Prediction with Large Language Model and Dual-task Learning' we do not include it previously since it is just published in August, 2025 ICJAI.
> > >
> > > | Model | Size | Accuracy (%) | Continue Pretraining | Tokenizer Mods | Training Data |
> > > |-------|------|--------------|-------------|----------------|---------------|
> > > | **Base LLM** | | | | | |
> > > | GPT-4o | - | 0.7 | No | No | - |
> > > | GPT-4.1-mini | - | 11.3 | No | No | - |
> > > | DeepSeek-V3-671B | 671B | 8.6 | No | No | - |
> > > | o4-mini | - | 12.2 | No | No | - |
> > > | DeepSeek-R1 | 671B | 11.2 | No | No | - |
> > > | Qwen2.5-0.5B | 0.5B | 0.0 | No | No | - |
> > > | **Finetuned LLM** | | | | | |
> > > | ChemDFM-v1.5-13B | 13B | 17.9 | Chemical | No | Extended |
> > > | LlaSMol-Mistral | - | 2.0 | Chemical | No | Extended |
> > > | RetroDFM-R-7B | 7B | 59.0 | Chemical | No | Extended |
> > > | ChemDual (Llama-3.1-8B) | 8B | 49.95 | No | Modified | 4.4M reactions |
> > > | **Ours-0.5B (without AAM)** | 0.5B | 69.5 | No | No | Standard USPTO |
> > > | **Ours-0.5B (with AAM)** | 0.5B | **74.8** | No | No | Standard USPTO |
> > >
> > > The evidence clearly shows that **larger models with extensive chemical pretraining and tokenizer modifications underperform our compact approach**. RetroDFM-7B not only fine-tunes but continues pretraining on large chemical datasets, yet achieves significantly lower accuracy (59.0% vs our 74.8%). Similarly, ChemDual employs an 8B model with explicit tokenizer modifications and trains on 4.4M reactions, yet achieves only 49.95% accuracy.
> > >
> > >
> > >
> > > ---
> > >
> > > ### **2. Tokenization and Format Understanding**
> > >
> > > We understand tokenization can directly impact performance, which is why we highlight that competing methods explicitly optimize this aspect. However, we disagree with the premise that LLMs perform poorly on reaction prediction primarily due to format misunderstanding.
> > >
> > > The poor performance of general-purpose LLMs supports our position:
> > > - **GPT-4o**: 0.7%
> > > - **DeepSeek-V3-671B**: 8.6%
> > > - **o4-mini**: 12.2%
> > > - **DeepSeek-R1**: 11.2%
> > >
> > > These models excel at competition-level mathematics, coding, and general science questions, yet perform poorly on chemical reaction prediction. This demonstrates that **chemical understanding requires specialized architectural design beyond general reasoning capabilities**. The post-training phase is crucial, but it's unfair to attribute fine-tuned LLM improvements in mathematics and coding solely to format learning while dismissing chemical prediction improvements similarly.
> > >
> > >
> > >
> > > ---
> > >
> > > ### **3. Core Contribution: Data-Centric Innovation**
> > > We show how the current LLM can work on reaction prediction but not using tokenization or large LLMs just by focusing on the data.
> > >
> > > Our fundamental contribution shifts the focus from scale-centric to data-centric approaches. While the field has emphasized larger models and more data, we demonstrate that:
> > >
> > > **The reaction prediction problem has specific structural characteristics** that general LLMs struggle to capture, requiring specialized solutions rather than just scale-based approaches.
> > > We improve with: **Better representations enable smaller models to outperform larger counterparts**.
> > > **Chemical intelligence should be architecturally embedded**.
> > >
> > > ---
> > >
> > > ### **4. Commitment to Revisions**
> > >
> > > We acknowledge the need to include recently published work and will expand our discussion to incorporate:
> > > - **ChemDual** and other dual-task learning approaches
> > > - **Detailed comparison** of tokenization strategies and their impacts
> > >
> > > We will also enhance the clarity and accessibility of our writing to ensure the broader audience can appreciate our contributions.
> > >
> > > ---

---

### Official Review · Reviewer_HTfX · 2025-11-01

**Soundness:** 3
**Presentation:** 3
**Contribution:** 3
**Rating:** 6
**Confidence:** 4

**Summary:**

The paper describes a unified framework for reaction prediction. It is based on three technical components: contrastive learning on mapped and unmapped data, bi-directional multi-task learning for forward reaction prediction and retrosynthetic analysis, and structured plan-based reasoning. The resulting model is compact and demonstrates 14% accuracy improvement over baselines.

**Strengths:**

AAM-0 contrastive learning appears useful, so does bi-directional training. After all, both forward reaction prediction and retrosynthetic analysis come down to determining a set of reactants, a set of products, and a set of condition (in the broad sense of the word).

**Weaknesses:**

Peculiar citation logic: retrosynthetic analysis citation points at 1969 paper, yet forward reaction prediction started only in 2019 according to the selected citation. Please do better.

It seems from the language of the paper, that the authors don't quite realize a fundamental difference between forward- and retrosynthetic tasks. The forward task starts with completely specified set of the reactants (along with conditions) and aims to predict the set of products along with yields. The retrosynthetic task starts with a single desirable product - this doesn't mean that it has to be the only product of the predicted reactions. This makes retrosynthesis a fundamentally underspecified task that is supposed to recover multiple sets of reactants producing different sets of products all containing target product with different yields. Retrosynthesis is fundamentally a prediction of ensembles of forward reactions, such that the intersection of their product sets has at least one common product (the target).

The prompt appears to be explicitly referring to known reaction types - it cannot be better in any sense than the approaches criticized in the introduction for combinatorial explosion. The only reason why the authors do not have combinatorial explosion of the length of their prompt is because they did not properly enumerate the reaction types.

The argument about "inefficient representation" is confusing. SMILES can be canonicalized (just like IUPAC names) for uniqueness, ambition to use SMILES to describe reactions led to the development of Reaction SMILES. Throw in SMARTs for the completeness of the discourse. These are chemoinformatic representations - the paper does nothing to improve them. Internal (latent and such) representations obtained by deep neural models are as numerous as deep neural models - the paper doesn't really describe any novelty. Using contrastive learning protocol does not change the nature of the representation, this contrastive training protocol appears to improve its quality which is often the case with contrastive learning.

There's only one task in one benchmark (out of eight tasks, two options re: reaction types,  and three benchmark) where 14% accuracy improvement is observed. In majority of the comparisons the gains are smaller. This is important considering that some baselines are already in +90% range.

**Questions:**

Please fix the references.

Please tighten the narrative and just address what you realistically accomplished (contrastive training protocol on mapped an unmapped SMILES is not a new representation; prompt referring to reaction types is an uncontrolled sample of a combinatorial set of reaction types, etc).

Please discuss the sensitivity of the outcomes to LLM prompting. This is the biggest source of the uncertainty in the framework.

As far as uncertainty goes, please at least discuss uncertainty propagation in this framework.

---

> ### Author Response · Authors · 2025-11-20
> **Experiment part.**
>
> **Reviewer HTfX Q1. Magnitude of the Claimed Improvement:**
> **Reviewer Comment:** "There's only one task in one benchmark... where 14% accuracy improvement is observed. In majority of the comparisons the gains are smaller."
>
> **Author Response:**
> Yes we agree the most significant improvement is in the top-1 accuracy (14%) in retrosynthesis.
> But this is important to show how the model learns from the datasets and validate our training pipelines. Though it is still far from the practical use in the real world.
> For top-K accuracy and forward prediction, ours still show consistent improvement.
>
> We add the evaluation of the base model (Qwen2.5-0.5B and Deepseek-V3/R1-671B, closed-source model like gpt-4.1-mini and o4-mini) without any fine-tuning on retrosynthesis tasks. Qwen2.5-0.5B achieved near-zero accuracy (approximately 0%), indicating no prior knowledge of chemical reactions. And even large model or closed-source model like Deepseek-V3/R1-671B,gpt-4.1-mini and o4-mini achieve around 10% top-1 acc on USPTO-50K. Also we compare with LLM based models like ChemDFM(13B) and retroDFM(7B) finetuned with chemistry data. Our 0.5B model also have better performance which validate our training pipeline.
>
> | Model                  | top-1 Accuracy (%) |
> |------------------------|--------------|
> | **Base LLM**           |              |
> | gpt-4o                 | 0.7          |
> | gpt-4.1-mini           | 11.3         |
> | DeepSeek-V3-671B       | 8.6          |
> | o4-mini               | 12.2         |
> | DeepSeek-R1            | 11.2         |
> | Qwen2.5-0.5B           | 0.0          |
> | **Finetuned LLM**      |              |
> | ChemDFM-v1.5-13B       | 17.9         |
> | LlaSMol-Mistral        | 2.0          |
> | RETRODFM-R-7B          | 59.0         |
> | Ours-0.5B (without AAM)| 69.5         |
> | Ours-0.5B (with AAM)   | **74.8**     |
>
> The results for the baselines are adopted from the RetroDFM paper.
> - Zhang S, Li H, Chen L, et al. Reasoning-Driven Retrosynthesis Prediction with Large Language Models via Reinforcement Learning[J]. arXiv preprint arXiv:2507.17448, 2025.
>
> **Reviewer HTfX Q2. Sensitivity to Prompting and Uncertainty Propagation:**
>
> **Reviewer Comment:** "Please discuss the sensitivity of the outcomes to LLM prompting. This is the biggest source of uncertainty... discuss uncertainty propagation."
>
> **Author Response:** These are excellent suggestions. We agree that prompt sensitivity is a common challenge for LLMs, particularly for compact models like our 0.5B parameter architecture fine-tuned with LoRA. Our framework incorporates a key design choice to mitigate this:
> - **Structured Output with Plan Tokens:** We enforce a structured generation process using explicit `<plan>` and `<answer>` tokens. This technique narrows the output space and reduces variability compared to free-form instructional prompting.
>
> | Model                  | top-1 Accuracy (%) |
> |------------------------|--------------|
> | Prompt                 | 74.8         |
> | Prompt'           | 73.2         |
> | Prompt' without \<plan\>       | 71.9          |
>
> Prompt means the original prompt. prompt' means the perturbed prompt by modifying a few sentences.
>
> - Another source of uncertainty comes from the temperature. top-1 predictions are obtained via greedy decoding (temperature = 0). For top-k predictions (k > 1), we set the temperature to 1.4 (optimal per our analysis) to sample multiple candidates (n=20). These candidates are then ranked by their log-probabilities, and the top-k are selected. This balances exploration and exploitation, with temperature 1.4 maximizing performance before degradation at higher values.

---

> ### Author Response · Authors · 2025-11-20
> **About tighten the narrative and reference**
>
> **Reviewer HTfX Q3. About the "AAM-0 Representation":**
>
> **Reviewer Comment:** "contrastive training protocol on mapped an unmapped SMILES is not a new representation... the paper doesn't really describe any novelty."
>
> **Author Response:**
> We agree that we do not propose any new chemoinformatic representations. Here we just propose a new data representation for the LLM to understand the reactions
> It is more from a data-centric view instead of a cheminformatics view.
>
> **Reviewer HTfX Q4. Conceptual Understanding of Forward vs. Retrosynthesis:**
>
> **Reviewer Comment:** "The authors don't quite realize a fundamental difference... Retrosynthesis is fundamentally a prediction of ensembles of forward reactions..."
>
> **Author Response:** We thank the reviewer for this insightful comment. We agree that retrosynthesis is inherently underspecified, as multiple reactant sets can lead to a target product, often with different by-products and yields.
>
> For the current work, we want to show how well a AI model can do to make forward and retrosynthesis predictions aligning with the benchmark datasets given the relatively low top-1 performance for the current baselines.
> We agree this setting and evaluation is not that realistic and practical in the real world. But we try to move towards it.
> Our goal in this work was to create a fair comparison with existing single-step prediction baselines, which typically treat it as a deterministic inverse problem to the datasets. However, we recognize that framing it this way oversimplifies the task. We will refine our problem formulation to explicitly acknowledge this fundamental difference in the limitation.
>
> **Reviewer HTfX Q5. Reaction Type Prompt and Combinatorial Explosion:**
>
> **Reviewer Comment:** "The prompt appears to be explicitly referring to known reaction types... The only reason why the authors do not have combinatorial explosion... is because they did not properly enumerate the reaction types."
>
> **Author Response:** We thank the reviewer for this important observation.  We wish to clarify our approach:
> - For retrosynthesis without types(Table 1 without reaction types), we do not include the reaction type in the model input.
> - The inclusion of a reaction type-known condition is primarily for direct and fair comparison with established baselines on standard benchmarks (e.g., USPTO-50k with/without reaction type), where this is a common evaluation setting.
>
> - We acknowledge that the common top-1 evaluation setting (predicting the single most likely outcome) is less practical for real-world retrosynthesis.
>
> **Reviewer HTfX Q6. Citation Logic:**
>
> **Reviewer Comment:** "Peculiar citation logic: retrosynthetic analysis citation points at 1969 paper, yet forward reaction prediction started only in 2019 according to the selected citation. Please do better."
>
> **Author Response:** This is a valid point. We acknowledge this creates an imbalanced perception of the fields' histories. We will revise it to provide a more balanced and accurate historical context for both tasks.

---

> ### Author Response · Authors · 2025-11-25
>
> We are following up on the response we provided. Please let us know if it addresses your questions, ​​or if you have any further ones—we are happy to explain.​

---

### Meta-Review · Area_Chair_z2sU · 2025-12-17

**Summary:**

This paper introduces a unified framework for chemical reaction prediction that integrates LLMs with contrastive learning and multi-task learning. While reviewers generally acknowledge the technical soundness of the proposed approach, several critical concerns remain unaddressed despite the authors' inclusion of detailed experiments and explanations. The major points are summarized as follows:

1. Limited methodological novelty: The proposed framework demonstrates competent integration of existing techniques, yet it falls short of offering substantial innovation from a methodological perspective. The conceptual advance over prior work remains unclear.

2. Potential data leakage issue: There is a serious concern regarding possible data leakage, as the base LLM may have been pre-trained on or exposed to the public evaluation datasets used in this study.

3. Inadequate related work and paper organization: The literature review is not sufficiently comprehensive or well-structured, failing to clearly position this work within the existing research landscape. Additionally, the overall organization of the manuscript requires substantial improvement to enhance clarity and logical flow.

**Reviewer Concerns:**

Regarding the concerns with inadequate related work and unclear paper organization, the authors have provides some suggestions for modification, which may help improve the quality of presentation. Still, the above two major concerns of limited methodological novelty and potential data leakage issues are not fully addressed in the rebuttal, which may require a lot of algorithmic innovation and more experimental evaluations.

**Reviewer Scores:**

I don't think they will change their scores, since they have stated clearly in the discussion that some issues have been addressed, but some critical problems remain untackled and require a lot of additional work.

---

### Decision · Program_Chairs · 2026-01-26

Reject